# Correlation Dependence between Hydrophobicity of Modified Bitumen and Water Saturation of Asphalt Concrete

**Antonina Dyuryagina, Yuliya Byzova \*, Kirill Ostrovnoy and Aida Lutsenko**

Department of Chemistry and Chemical Technology, Manash Kozybayev North Kazakhstan University, Pushkin Str. 86, Petropavlovsk 150000, Kazakhstan; adyuryagina@inbox.ru (A.D.); kostrovnoy@mail.ru (K.O.); l-a.13@mail.ru (A.L.)
**\*** Correspondence: yuliyabyzovva@gmail.com; Tel.: +7-7752359923

**Abstract:** Improving the durability of asphalt concrete road surfaces by increasing their moisture resistance is an urgent task. Modified bituminous binders should be compacted into coatings with the lowest possible water saturation. The purpose of this study was to establish the effect of modifiers on the hydrophobicity of bituminous films in order to achieve minimum water saturation and to build a mathematical model of the wetting process with water. As modifiers, we used a product of amination of distillation residues of petrochemistry, waste sealing liquid (a solution of high molecular weight polyisobutylene in mineral oil), and a condensation product of polyamines and higher fatty acids. The water-repellent effect of modifiers was studied by measuring the contact angle of bituminous film with a water drop. The water saturation of asphalt concrete samples was determined by the amount of water absorbed by asphalt concrete at 20 °C. A close correlation was revealed between the hydrophobicity of modified bitumen and the water saturation of asphalt concrete. Generalized equations and a graphical representation of a function of several variables allowed for optimizing compositions by the content of modifiers to achieve the required performance properties of asphalt concrete coatings.

**Keywords:** water saturation; bitumen films; asphalt concrete; hydrophobicity; modified bitumen; comparative study; methods development





## 1. Introduction

At present, increasing the durability of road asphalt concrete pavements is an urgent task, the solution of which provides a significant economic effect, achieved by increasing the time between repairs as well as the overall service life of roads [1–3].

The decrease in the strength and stability of road asphalt concrete pavement and the formation of rutting and other defects are caused by the deformation effects of the environment, such as high humidity and seasonal temperature changes [4–6]. The upper layers of road surfaces are regularly exposed to moisture, which leads to the increased water saturation of asphalt concrete. With prolonged exposure, water penetrates into the pores of asphalt concrete, saturates bitumen, and penetrates through the defective places of bitumen layers to the surface of mineral aggregates. Even more destructive is the effect of water freezing in the pores of asphalt concrete or in the pores of the stone material contained in it. Freezing water, increasing in volume, causes a large amount of stress on the walls of the pores, resulting in micro-cracks that fill with thawing water. These factors contribute to the weakening of structural bonds in asphalt concrete, which leads to the peeling of bituminous films and the destruction of the coating under the influence of vehicles [7–13].

Modified bituminous binders should be compacted into coatings with the lowest possible water saturation. The development of modified bitumen-mineral compositions that can increase the life of pavements in a sharply continental climate (seasonal temperature fluctuations from −40 °C to +40 °C and humidity) with a slight increase in cost is undoubtedly a strategically important task [14–16].

Numerous studies [17–25] devoted to the modification of bitumen binders, in order to increase the water resistance of bitumen, are developing mainly in two directions. The first direction is associated with the introduction of amphiphilic compounds of various molecular weight composition into the bitumen composition, which act as adhesive additives that improve the adhesion of bitumen to the surface of the mineral filler. In the role of surface-active substances (surfactants) in asphalt concrete compositions, cationic surfactants, diamines, and polyamines were initially used [26–29]. In the United States and China, additives such as amines and ammonium salts are widely used [30,31]. In the last decade, nonionic surfactants (esters of polyoxyethylene alkyphenols OP-7 and OP-10), which are most sensitive to both alkaline and acidic mineral materials, have become widespread [32,33].

The second direction of research is focused on the introduction of polymer additives, which significantly improve the structural-mechanical and waterproofing characteristics of asphalt concrete due to the formation of a spatial network in the volume of the bituminous binder phase [17,18,20,22,34–37]. The effect of increasing water resistance is due to a slowdown in the diffusion of water through a film of bitumen-polymer binders. Currently, the following types of polymers have found application as a polymeric bitumen modifier: elastomers: SB (styrene-butadiene copolymer), SBS (styrene-butadiene-styrene) [28,37], and NBR (butadiene-nitrite rubber) [38]; thermoplastics: EVA (ethyl vinyl acetate) [39,40], EMA (ethylene methyl acrylate) [24,41], APP (atactic polypropylene) [42], and PE (polyethylene) [22,24]; and thermosetting resins [24]. The compatibility of bitumen with thermoplastic elastomers (DST-30, IST-30), butyl rubber (BR-2045T), and ethylene-propylene rubber (SKEP M-40 and M-60) has been studied. It has been shown that stable polymer bitumen compositions can be obtained using mixtures with paraffin-naphthenic oils [43]. It has been proven that the efficiency of bitumen modification in each individual case depends on the quantitative ratios of the polymer and bitumen, their compatibility, as well as the temperature regimes for the preparation of the modified bitumen binder [24,44].

While in the first direction of scientific research, surface phenomena occurring at the interfacial boundary "bitumen-mineral filler" are dominant, processes in the bulk phase of the bituminous binder in the second direction prevail. Rather less attention is paid to the bitumen-air interface, although, in essence, it is the first barrier that prevents the penetration of water into the pavement. By varying the type and concentration of additives, it is possible to obtain composite materials with higher water-repellent properties of the bitumen film as a result of a change in the interfacial layer at the boundary with air.

One of the promising ways to solve the problem of modifying asphalt concrete is the partial replacement of expensive additives with cheaper waste. Various types of polymer waste can be used in the production of road building materials. The production of modified asphalt mixes using recycled materials helps to reduce the large amount of waste generated from various sources. It also reduces the consumption of fossil materials, therefore minimizing the impact of the road industry on the environment [45–48].

The purpose of this study is to establish the effect of modifiers of various nature (AG-4I, AC-1 and AMDOR-10) on the hydrophobicity of bituminous films in order to achieve minimum water saturation and build a mathematical model of the wetting process with water.

To achieve the goal, the following tasks were set:

1. To study the influence of the concentration of additives in bitumen on the processes of the water wetting of bitumen films and the surface tension of binary "bitumen-additive" and ternary "bitumen-polymer-additive" systems.
2. To develop mathematical models of the processes of water wetting of modified bitumen films, based on the established physical and chemical laws.
3. To determine the effectiveness of additives in terms of the water saturation of modified asphalt concrete pavements.

4.  To derive correlations between the contact angle of wetting bitumen films with water and the water saturation index of asphalt concrete samples, which makes it possible to predict the hydrophobicity of the formed asphalt concrete coatings.

## 2. Materials and Methods

### 2.1. Materials and Laboratory Tests

As mineral fillers in the preparation of asphalt concrete mixtures, crushed stone of several fractions, crushed sand from the "Gora Zmeevaya" gabbro deposit (Revda, Russia), as well as activated mineral powder from carbonate rocks MP-1 produced by PromStroyDekor LLC (Nevyansk, Russia) were used. As a binder, was used the original and modified bitumen grade BND 100/130 of Ural Bitumen Terminal Ltd. (Ekaterinburg, Russia). As a prototype, the composition of hot dense asphalt concrete type A grade I, as the most widely used in the climatic zone of Northern Kazakhstan, was taken.

The composition of the crushed stone-mastic asphalt concrete mixture was selected empirically. The recipe is based on the requirements of [49]. When selecting the composition of mixtures, we proceeded using the conditions for achieving the necessary physical and mechanical characteristics of asphalt concrete at the highest density and the lowest consumption of bituminous binder. The designed composition of the hot asphalt concrete sample used for further research is presented in Table 1.

**Table 1.** Quantitative composition of the asphalt mix sample.

| Component Name | Content, % wt. |
| --- | --- |
| Crushed stone fr. 11.2–16 mm | 25.0 |
| Crushed stone fr. 8–11.2 mm | 12.0 |
| Crushed stone fr. 4–8 mm | 20.0 |
| Crushed sand 0–4 mm | 41.0 |
| Mineral powder | 2.0 |
| Bitumen binder content, % by weight of the mineral part | 4.6 |

To determine the properties of asphalt concrete, cylindrical samples (Figure 1) were used. The molding of the samples was carried out in a metal mold with liners at a temperature of 90–100 °C. The samples were compacted on a press at a pressure of 40 MPa for 3 min.

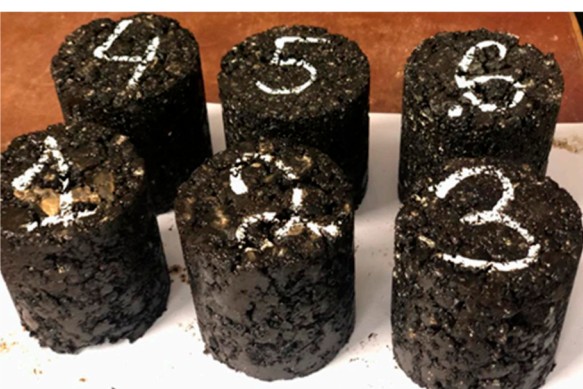

**Figure 1.** Samples of asphalt concrete mix.

Studies of the physical and mechanical characteristics of the obtained unmodified samples of asphalt concrete mixtures were carried out in accordance with the specifications of [50]. The physical and mechanical properties of asphalt concrete are shown in Table 2. It was found that the asphalt concrete mixture without the introduction of modifiers did not meet the requirements of GOST in terms of water saturation.

**Table 2.** Test results for the unmodified asphalt mix.

| The Name of Indicators | Requirements [50] | Actual Performance |
|---|---|---|
| Average density, g/cm$^3$ | Not standardized | 2.69 |
| Porosity of the mineral framework, % by volume | 14 to 19 | 15.7 |
| Residual porosity, % by volume | 2.5 to 5.0 | 3.85 |
| Water saturation, % by volume | 2.0 to 5.0 | 7.9 |
| Compressive strength, MPa | | |
| 20 °C | At least 2.5 | 4.8 |
| 50 °C | At least 1.0 | 1.8 |
| 0 °C | No more than 11 | 10.7 |
| Water resistance | No less than 0.90 | 0.91 |
| Water resistance with long-term water saturation | No less than 0.85 | 0.86 |
| Adhesion of bitumen with the mineral part of the mixture | Must be provided | Provided |
| Crack resistance—tensile strength at a split at a temperature of 0 °C, MPa | 3.5 to 6.0 | 5.4 |
| Coefficient of internal friction | No less than 0.87 | 0.92 |
| Shear adhesion at 50 °C, MPa | No less than 0.25 | 0.55 |

For modification of the bituminous binder, we used the following:

- AC-1: A product of amination of KON-92 and distillation residues of petrochemistry (molecular weight: 250 a.m.u) [51], which is a mixture of primary and secondary amines of the general formula: R′-NH2, R′-NH-R″, where R′ is n-butyl, and R″ is 2-ethyl-2-hexenyl in a ratio of 1:3.
- AG-4I: A waste sealing liquid, solution of high molecular weight polyisobutylene (PIB) in mineral oil (molecular weight: 5400 a.m.u), manufacturer SIF "Germika" (Moscow, Russia).
- AMDOR-10: A condensation product of polyamines and higher fatty acids (molecular weight: 2260 a.m.u), manufactured by Amdor CJSC (Saint-Petersburg, Russia).

*2.2. Preparation of Modified Bituminous Compositions*

To prepare the modified bituminous binder, a porcelain container, a thermometer, a laboratory mechanical mixer, and electric heating were used. Initially, the bitumen was heated to a mobile state (80 °C) to remove moisture and air bubbles, and then, the temperature was raised to 130 °C, after which the binder was held for 30 min. After the specified time, the modifiers were dosed. The consumption of additives varied from 0 to 2.0 g/dm$^3$. The bitumen-additive binary composition was stirred at a fixed temperature (130 °C) for 40 min. The optimal mixing time, which makes it possible to achieve equilibrium indicators of the specific surface energy $\sigma_{l\text{-}g}$ and the wetting angle θ of modified bitumen films with water, was established from the results of preliminary kinetic studies. To prepare the "bitumen-polymer-surfactant" ternary compositions, a sealing liquid solution ($C_{AG\text{-}4I} = 0.5 \div 2.0$ g/dm$^3$) was added into the molten bitumen, and the composition was stirred in a thermostatically controlled mode (T = 130 °C) for 40 min. Then, AC-1 ($C_{AC\text{-}1} = 0.5 \div 2.0$ g/dm$^3$) was dosed into the «bitumen-AG-4I» binary system with a fixed sealing liquid content, maintaining the composition at constant stirring and at the same temperature for another 40 min to achieve an equilibrium system state.

*2.3. Method for Applying Bituminous Compositions to the Surface of a Glass Plate*

The application of bituminous compositions on the surface of a glass plate was carried out by pouring, which consisted of applying a bituminous solution flowing down onto a plate previously cleaned with acetone, which was in a horizontal position. Then, the plate was rotated by 45° to remove excess bitumen, then placed horizontally, and cooled at 20 °C.

### 2.4. Method for Determining the Hydrophobizing Effect of Modifiers

The water-repellent effect of modifiers was studied by measuring the contact angle of wetting a bituminous film deposited on the surface of a glass plate with a water drop on an automatic installation of the ACAM series.

A drop of water was applied to the surface of the bitumen film using automatic feeding through a dosing syringe, and by injecting and pumping out the liquid, the optimal drop volume was achieved at a fixed drop rate from the syringe, setting the necessary program parameters. A drop of a given volume was transferred to the surface of the substrate by raising the object table, on which the bitumen plate was located (Figure 2).

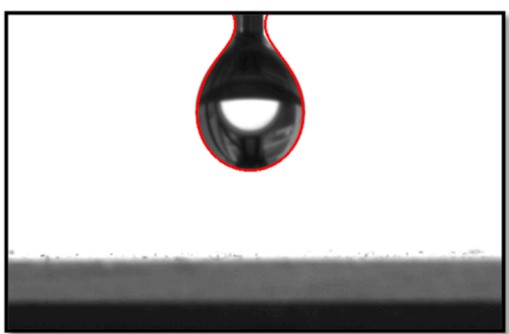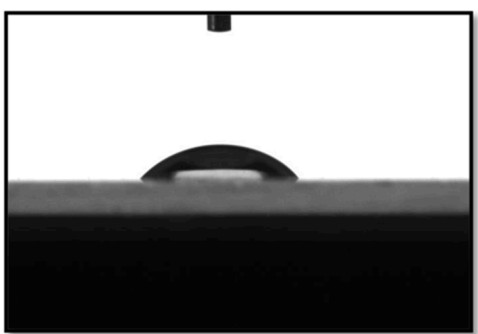

**Figure 2.** The process of transferring a drop of solution from the tip of the needle of the dosing syringe onto the surface of the substrate.

Static contact angles at a steady state were measured 20 s after dosing. The contact angle was determined by five parallel measurements. The measurement accuracy was ±0.05°.

### 2.5. Method for Determining the Specific Surface Energy at the "Liquid-Air" Interface

The measurement of surface tension was carried out by the hanging drop method on an automatic installation of the ACAM series, forming a drop using a dosing syringe by maximum dosing of the bituminous composition until the moment of detachment. The fixed image of a hanging drop was analyzed along with the profile contour using the Young-Laplace equation with the Apex Acam Software (version 2.026.088.1).

The automatic calculation of surface tension was performed for several hundred pairs of points in less than one second. Such a rate made it possible to trace the change in surface tension (non-equilibrium surface tension) over relatively short periods of time. Surface tension was determined by five parallel measurements.

### 2.6. Method for Determining the Water Saturation of Asphalt Concrete Samples

Water saturation indicators were determined according to [50]. The water saturation of asphalt concrete samples was determined by the amount of water absorbed by asphalt concrete at 20 °C. The rapid water saturation of asphalt concrete was carried out by creating a vacuum above the surface of the water in which the samples were immersed.

Samples of asphalt concrete were placed in a vessel with water, the temperature of which was 20 °C. The water level above the samples was set at a constant 3 cm. The vessel with the samples was installed in a vacuum drying oven, where a residual pressure of 2000 Pa was created and maintained for 1 h and 30 min. Then, the pressure was brought to an atmospheric level, and the samples were kept in the same vessel with water at a temperature of 20 °C for 1 h. After that, the samples were removed from the water, wiped with filter paper, and weighed with an error of 0.01 g in air and in water. The increase in the mass of the sample corresponded to the amount of water absorbed by the sample. The increment in the mass of the sample, referred to as the initial volume of the sample, is its water saturation by volume.

### 2.7. Method of Probabilistic-Deterministic Planning

Modeling the impact of modifiers on the processes of wetting bitumen films with water and the water saturation of asphalt concrete samples was carried out within the framework of the probabilistic–deterministic planning (PDP) method. Research work using PDP consisted of several stages:

1.  Determining factors and levels of their variation.
2.  Constructing an experiment plan in the form of a plan matrix consisting of *m* rows corresponding to the number of levels of factor $x_1$, and n columns, the number of levels of factor $x_2$.
3.  Conducting an active experiment according to the generated plan matrix and establishing the numerical values of the response function (output parameter).
4.  Sampling the response function for each level of each factor.
5.  Constructing partial dependencies of the response function on each factor.
6.  Approximating partial dependencies and the derivation of a generalized mathematical model.

The main factors (input parameters) were determined to be the content of the surfactant AC-1 ($C_{AC-1}$, g/dm$^3$: 0–2.0) and the content of the polymer additive AG-4I ($C_{AG-4I}$, g/dm$^3$: 0–2.0) in the bituminous binder.

The numerical values of the levels for each factor are presented in Table 3.

**Table 3.** Numerical values of levels for each factor.

| Factors | Level | | | | |
|---|---|---|---|---|---|
| | **1** | **2** | **3** | **4** | **5** |
| $C_{AC-1}$, g/dm$^3$, ($x_1$) | 0 | 0.5 | 1.0 | 1.5 | 2.0 |
| $C_{AG-4I}$, g/dm$^3$, ($x_2$) | 0 | 0.5 | 1.0 | 1.5 | 2.0 |

The values of the contact angle of wetting ($\theta$) and of the water saturation index (W), which were determined experimentally, were taken as the response function.

Experiments were carried out according to the constructed orthogonal, multilevel plan-matrix of a two-factor experiment (Table 4).

**Table 4.** The multi-level plan-matrix of a two-factor experiment.

| Factor Levels $x_1$ | Factor Levels $x_2$ | | | | |
|---|---|---|---|---|---|
| | **1** | **2** | **3** | **4** | **5** |
| 1 | $y_1$ | $y_6$ | $y_{11}$ | $y_{16}$ | $y_{21}$ |
| 2 | $y_2$ | $y_7$ | $y_{12}$ | $y_{17}$ | $y_{22}$ |
| 3 | $y_3$ | $y_8$ | $y_{13}$ | $y_{18}$ | $y_{23}$ |
| 4 | $y_4$ | $y_9$ | $y_{14}$ | $y_{19}$ | $y_{24}$ |
| 5 | $y_5$ | $y_{10}$ | $y_{15}$ | $y_{20}$ | $y_{25}$ |

After the implementation of the active experiment (Table 4), the experimental data array was sampled for each level of each factor according to Table 5.

**Table 5.** The sampling of experimental data array for each level of each factor.

| Factor Levels $x_1$ $C_{AC-1}$, g/dm$^3$ | Sample | Factor Levels $x_2$ $C_{AG-4I}$, g/dm$^3$ | Sample |
|---|---|---|---|
| 0 | $(y_1 + y_6 + y_{11} + y_{16} + y_{21})/5$ | 0 | $(y_1 + y_2 + y_3 + y_4 + y_5)/5$ |
| 0.5 | $(y_2 + y_7 + y_{12} + y_{17} + y_{22})/5$ | 0.5 | $(y_6 + y_7 + y_8 + y_9 + y_{10})/5$ |
| 1.0 | $(y_3 + y_8 + y_{13} + y_{18} + y_{23})/5$ | 1.0 | $(y_{11} + y_{12} + y_{13} + y_{14} + y_{15})/5$ |
| 1.5 | $(y_4 + y_9 + y_{14} + y_{19} + y_{24})/5$ | 1.5 | $(y_{16} + y_{17} + y_{18} + y_{19} + y_{20})/5$ |
| 2.0 | $(y_5 + y_{10} + y_{15} + y_{20} + y_{25})/5$ | 2.0 | $(y_{21} + y_{22} + y_{23} + y_{24} + y_{25})/5$ |

Based on the sampling of the experimental data array (Table 5), partial dependencies of the response functions on the content of modifiers were plotted.

In the last stage, partial dependencies were approximated to obtain one-parameter equations characterizing the influence of each factor separately on the response function. To build a multifactorial statistical mathematical model (generalized equation), we used the formula proposed by Protodyakonov, which in the case of a two-factor experiment takes the form:

$$y = \frac{f(x_1) \cdot f(x_2)}{y_{av}}, \tag{1}$$

where $f(x_1)$, $f(x_2)$ are one-parameter equations characterizing the influence of factors $x_1$ and $x_2$, respectively; $y_{av}$ is the average value of the actual value of the output parameter for all experiments (general average).

The values of $y_{av}$ were calculated by the formula:

$$y_{av} = \frac{\sum y_i}{n}, \tag{2}$$

where $\sum y_i$ is a set of experimental data in a matrix; $n$ is the total number of experiments in the plan matrix.

The estimation of the accuracy of the obtained mathematical models (generalized equations) was evaluated using the coefficients of nonlinear multiple correlation (R) and significance ($t_R$), which were calculated using Equations (3) and (4):

$$R = \sqrt{1 - \frac{(n-2)\sum(y_{ex} - y_{cl})^2}{(n-1)\sum(y_{ex} - y_{av})^2}}, \tag{3}$$

where n is the number of experiments; $y_{ex}$ is the experimental value of the response function; $y_{cl}$ is the calculated value of the response function; $y_{av}$ is the general average of experimental values of the response function.

$$t_R = \frac{R\sqrt{(n-2)}}{1 - R^2}, \tag{4}$$

## 3. Results

### 3.1. Binary Systems "Bitumen-Additive"

Table 6 shows values of contact angles of the wetting of bitumen film with water and surface tension in binary systems "bitumen-additive".

**Table 6.** Contact angle of wetting of bitumen film with water and surface tension in binary systems "bitumen-additive".

| AG-4I, g/dm$^3$ | $\theta$, ° | $\sigma$, mN/m | AC-1, g/dm$^3$ | $\theta$, ° | $\sigma$, mN/m | AMDOR-10, g/dm$^3$ | $\theta$, ° | $\sigma$, mN/m |
|---|---|---|---|---|---|---|---|---|
| 0 | 95.05 | 45.39 | 0 | 95.05 | 45.39 | 0 | 95.05 | 45.39 |
| 0.5 | 101.68 | 43.35 | 0.5 | 100.39 | 42.81 | 0.5 | 98.81 | 44.37 |
| 1.0 | 107.30 | 41.31 | 1.0 | 103.21 | 40.80 | 1.0 | 101.81 | 44.71 |
| 1.5 | 102.36 | 43.52 | 1.5 | 103.55 | 40.92 | 1.5 | 101.70 | 44.90 |
| 2.0 | 96.15 | 45.56 | 2.0 | 103.86 | 41.00 | 2.0 | 101.75 | 45.22 |

The results of physical and chemical studies of modified bitumen compositions (Table 6) were confirmed by water saturation indicators (W, %) of asphalt concrete samples, presented in Table 7.

**Table 7.** Cosine of the contact angle of the bitumen film with water and water saturation of asphalt concrete samples.

| AG-4I, g/dm³ | W, % | cos θ | AC-1, g/dm³ | W, % | cos θ | AMDOR-10, g/dm³ | W, % | cos θ |
|---|---|---|---|---|---|---|---|---|
| 0 | 7.87 | −0.09 | 0 | 7.87 | −0.09 | 0 | 7.87 | −0.09 |
| 0.5 | 5.65 | −0.20 | 0.5 | 6.05 | −0.18 | 0.5 | 6.30 | −0.15 |
| 1.0 | 4.02 | −0.30 | 1.0 | 4.85 | −0.22 | 1.0 | 4.92 | −0.21 |
| 1.5 | 4.80 | −0.21 | 1.5 | 4.90 | −0.23 | 1.5 | 5.20 | −0.20 |
| 2.0 | 5.20 | −0.11 | 2.0 | 5.05 | −0.24 | 2.0 | 5.85 | −0.20 |

### 3.2. Ternary Systems "Bitumen-AG-4I-AC-1"

The results of experimental measurements of the contact angle of the wetting of bitumen with water $\theta$ and surface tension $\sigma_{l\text{-}g}$ in mixed composition, including the presence of AG-4I and AC-1, are presented in Table 8.

**Table 8.** Contact angle of the bitumen film with water and surface tension in the triple systems "bitumen–AG-4I–AC-1".

| AG-4I, g/dm³ | AC-1, g/dm³ | θ, ° | σ, mN/m | AC-1, g/dm³ | θ, ° | σ, mN/m | AC-1, g/dm³ | θ, ° | σ, mN/m | AC-1, g/dm³ | θ, ° | σ, mN/m | AC-1, g/dm³ | θ, ° | σ, mN/m |
|---|---|---|---|---|---|---|---|---|---|---|---|---|---|---|---|
| 0 | 0 | 95.05 | 45.39 | 0.5 | 100.39 | 42.81 | 1.0 | 103.21 | 40.80 | 1.5 | 103.55 | 40.92 | 2.0 | 103.86 | 41.00 |
| 0.5 | 0 | 101.68 | 43.35 | 0.5 | 104.09 | 41.20 | 1.0 | 106.16 | 38.20 | 1.5 | 106.55 | 38.75 | 2.0 | 105.51 | 39.40 |
| 1.0 | 0 | 107.30 | 41.31 | 0.5 | 110.91 | 38.50 | 1.0 | 115.9 | 38.50 | 1.5 | 113.63 | 37.00 | 2.0 | 113.11 | 37.30 |
| 1.5 | 0 | 107.50 | 43.52 | 0.5 | 109.23 | 39.20 | 1.0 | 111.93 | 37.50 | 1.5 | 111.84 | 39.20 | 2.0 | 111.55 | 38.20 |
| 2.0 | 0 | 107.82 | 45.56 | 0.5 | 105.51 | 42.50 | 1.0 | 111.53 | 40.50 | 1.5 | 111.45 | 40.70 | 2.0 | 110.93 | 40.90 |

The calculated contact angle ($\theta_c$) was found as an additive value:

$$\theta_c = \theta_0 - (\Delta\theta_{AG-4I} + \Delta\theta_{AC-1}), \tag{5}$$

where $\Delta\theta_{AG-4I}$ and $\Delta\theta_{AC-1}$ are the change in the contact angle in the binary system relative to unmodified bitumen.

For a comparative assessment of these indicators, we determined $\Delta$:

$$\Delta = \theta_c - \theta_{ex} \tag{6}$$

The data obtained are presented in Table 9.

**Table 9.** Contact angle of the wetting of the bitumen film with water in the ternary systems "bitumen–AG-4I–AC-1".

| AC-1, g/dm³ | AG-4I, g/dm³ | $\Delta\theta_{AC-1}$, ° | $\Delta\theta_{AG-4I}$, ° | θc, ° | θex, ° | Δ, ° |
|---|---|---|---|---|---|---|
| 0.5 | 0.5 | 5.34 | 6.63 | 107.02 | 104.09 | +2.93 |
| 1.0 | 0.5 | 8.16 | 6.63 | 109.84 | 106.16 | +3.68 |
| 1.5 | 0.5 | 8.5 | 6.63 | 110.18 | 106.55 | +3.63 |
| 2.0 | 0.5 | 8.81 | 6.63 | 110.49 | 105.51 | +4.98 |
| 0.5 | 1.0 | 5.34 | 12.25 | 112.64 | 110.91 | +1.73 |
| 1.0 | 1.0 | 8.16 | 12.25 | 115.46 | 115.90 | −0.44 |
| 1.5 | 1.0 | 8.50 | 12.25 | 115.80 | 113.63 | 2.17 |
| 2.0 | 1.0 | 8.81 | 12.25 | 116.11 | 113.11 | +3.00 |
| 0.5 | 1.5 | 5.34 | 7.31 | 107.7 | 109.23 | −1.53 |
| 1.0 | 1.5 | 8.16 | 7.31 | 110.52 | 111.93 | −1.41 |
| 1.5 | 1.5 | 8.50 | 7.31 | 110.86 | 111.84 | −0.98 |
| 2.0 | 1.5 | 8.81 | 7.31 | 111.17 | 111.65 | −0.48 |
| 0.5 | 2.0 | 5.34 | 1.10 | 101.49 | 105.51 | −4.02 |
| 1.0 | 2.0 | 8.16 | 1.10 | 104.31 | 111.53 | −7.22 |
| 1.5 | 2.0 | 8.50 | 1.10 | 104.65 | 111.45 | −6.80 |
| 2.0 | 2.0 | 8.81 | 1.10 | 104.96 | 110.93 | −5.97 |

Dependencies of the water saturation index of asphalt concrete compositions with a fixed content of AG-4I (0, 0.5, 1.0, and 2.0 g/dm³) on the concentration of AC-1 are shown in Figure 3.

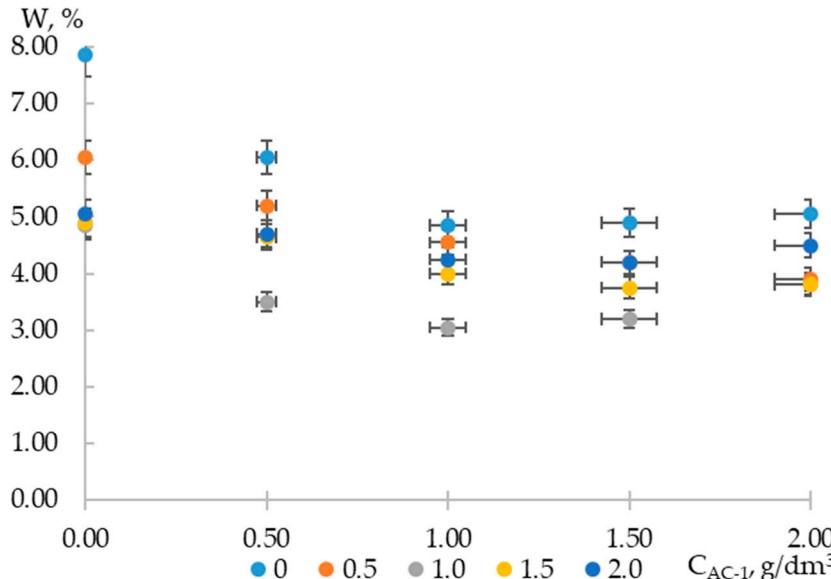

**Figure 3.** Dependencies of the water saturation index of asphalt concrete compositions with a fixed content of AG-4I (0, 0.5, 1.0, and 2.0 g/dm³) on the concentration of AC-1.

Dependencies of the cosine of the contact angle of the wetting of bitumen films with a fixed content of AG-4I (0, 0.5, 1.0, and 2.0 g/dm³) on the concentration of AC-1 are shown in Figure 4.

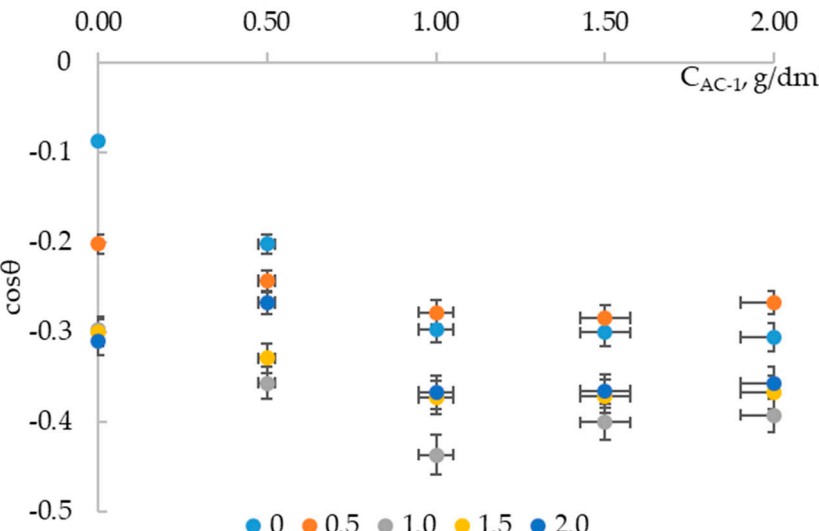

**Figure 4.** Dependencies of the cosine of the contact angle of the wetting of bitumen films with a fixed content of AG-4I (0, 0.5, 1.0, and 2.0 g/dm³) on the concentration of AC-1.

### 3.3. Results of Probability-Deterministic Modeling

The results of an active experiment to determine the wetting index of bituminous films with water and the water saturation of asphalt concrete pavements are presented in Table 10.

**Table 10.** Multilevel plan-matrix of a two-factor experiment.

| $C_{AC-1}$, g/dm$^3$ | $C_{AG-4I}$, g/dm$^3$ | | | | | | | | | |
|---|---|---|---|---|---|---|---|---|---|---|
| | 0 | | 0.5 | | 1.0 | | 1.5 | | 2.0 | |
| | $\theta$, ° | W, % | $\theta$, ° | W, % | $\theta$, ° | W, % | $\theta$, ° | W, % | $\theta$, ° | W, % |
| 0 | 95.05 | 7.87 | 101.68 | 5.65 | 107.30 | 4.02 | 107.50 | 4.80 | 107.82 | 5.20 |
| 0.5 | 100.39 | 6.05 | 104.09 | 5.20 | 110.91 | 3.50 | 109.23 | 4.65 | 105.51 | 4.70 |
| 1.0 | 103.21 | 4.85 | 106.16 | 4.55 | 115.90 | 3.05 | 111.93 | 4.00 | 111.53 | 4.25 |
| 1.5 | 103.55 | 4.90 | 106.55 | 4.20 | 113.63 | 3.20 | 111.84 | 3.75 | 111.45 | 4.20 |
| 2.0 | 103.86 | 5.05 | 105.51 | 3.90 | 113.11 | 3.82 | 111.65 | 3.80 | 110.93 | 4.50 |

A sampling of values for the contact angle of wetting and water saturation index by the levels of factors used is presented in Table 11.

**Table 11.** Sample of response functions for each level of concentration factors of two additives.

| $C_{AC-1}$, g/dm$^3$ | $\theta$, ° | W, % | $C_{AG-4I}$, g/dm$^3$ | $\theta$, ° | W, % |
|---|---|---|---|---|---|
| 0.0 | 103.87 | 5.51 | 0.0 | 101.21 | 5.74 |
| 0.5 | 106.03 | 4.82 | 0.5 | 104.80 | 4.70 |
| 1.0 | 109.75 | 4.14 | 1.0 | 112.17 | 3.52 |
| 1.5 | 109.40 | 4.05 | 1.5 | 110.43 | 4.20 |
| 2.0 | 109.01 | 4.21 | 2.0 | 109.45 | 4.57 |

Partial dependencies of the response functions on the content of AC-1 and AG-4I additives are shown in Figure 5.

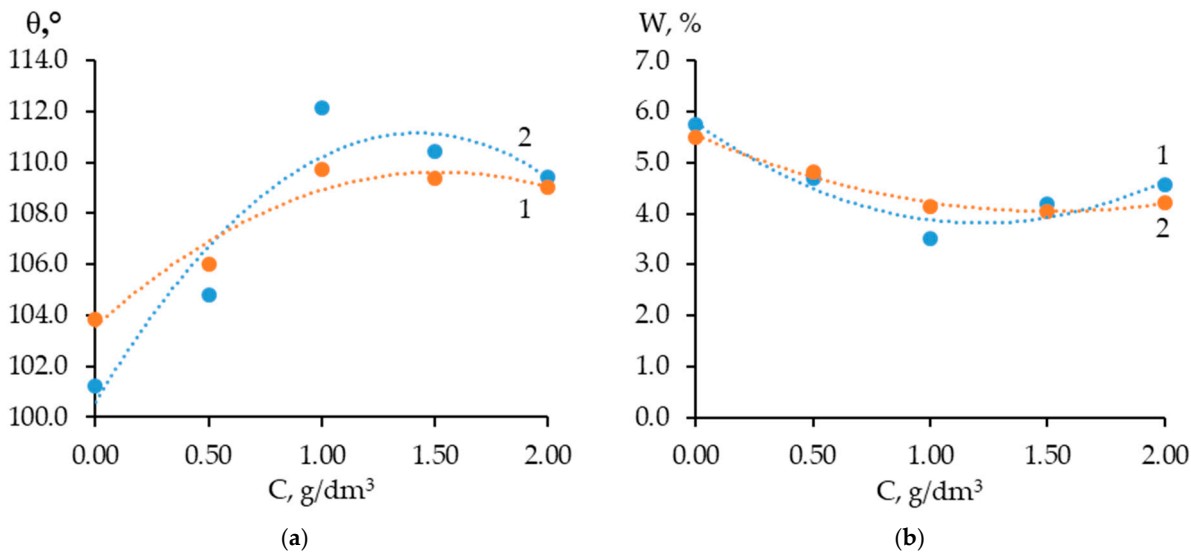

**Figure 5.** Partial dependencies of the contact angle (**a**) and the water saturation index (**b**) on the content of AC-1 and AG-4I: 1—AC-1; 2—AG-4I.

## 4. Discussion

### 4.1. Binary Systems "Bitumen-Additive"

As evidenced by the experimental data presented in Table 6, in the absence of additives, the bitumen surface ($\theta$ = 95.05°) was in the border region between hydrophilicity and hydrophobicity.

The addition of a limited concentration (C ≤ 1 g/dm$^3$) of a sealing liquid into bitumen led to an increase in the contact angle of wetting $\theta$. Thus, at C = 1 g/dm$^3$ of the modifier of polymeric nature, the value of $\theta$ increased by 12.25° relative to the base case and amounted to 107.30°. However, with a higher concentration of AG-4I, the opposite effect was recorded, a decrease in the contact angle of wetting up to 96.15° at C = 2 g/dm$^3$.

It should be noted that the concentration (C = 1 g/dm$^3$), which ensured the maximum hydrophobic effect, was also preserved in the presence of two other modifiers (AMDOR-10 and AC-1). With the same quantitative content of additives of amphiphilic nature, the θ values increased by 8.16° (AC-1) and 6.76° (AMDOR-10) relative to unmodified bitumen and amounted to 103.21° and 101.81°, respectively. Thus, their hydrophobic effect turned out to be 1.5–1.8 times lower than that of AG-4I. Another distinguishing feature is the stabilization of the values of the contact angle of wetting outside the indicated concentration range (C > 1 g/dm$^3$).

The observed changes in the contact angle of wetting with water of the modified bitumen coatings were consistent with the isotherms of surface tension ($\sigma_{l\text{-}g}$) and at the same time found their explanation from the standpoint of the formation of a surfactant adsorption layer on the "bitumen-air" interface. Judging by the isotherms of the surface tension of binary systems "bitumen-additive" in the concentration range of additives C ≤ 1 g/dm$^3$, the presence of diphilic molecules in the bulk phase of bitumen stimulated their concentration in the surface layer, which was confirmed by a decrease in the values of surface tension (Table 6).

In accordance with the Rebinder polarity equalization rule [52] on the interfacial surface, the surfactant molecules were oriented by the polar group into bitumen, and the hydrocarbon radicals turned out to be facing the air. This structure of the adsorption layer created a hydrophobic effect, which in turn was controlled by the length, branching of the hydrocarbon radical, and the number of hydrocarbon chains [53]. This clearly demonstrated the comparison of $\sigma_{l\text{-}g}$ and cos θ in isoconcentration (C = 1 g/dm$^3$) binary systems, "bitumen-AG-4I" and "bitumen-AC-1". With a similar decrease in surface tension (up to 41.31–40.80 mN/m), which was provided by the polar groups of surfactants, the values of *θ* increased in the system "bitumen-AG-4I" by 12.25°, and in the composition with AC-1, they only increased by 8.16°, which indicated a different degree of coverage of the interfacial surface "bitumen-air" by hydrophobic regions of surfactant molecules. The resulting adsorption layer in the bitumen composition with AMDOR-10 had an even lower degree of surface filling, judging by the slight change in $\sigma_{l\text{-}g}$ (0.68 mN/m) and the minimum increase in the contact angle (Δθ = 6.76°) in the studied series of additives.

The results of physical and chemical studies of modified bitumen compositions (Table 6) were confirmed by water saturation indicators (W, %) of asphalt concrete samples (Table 7).

During the tests, it was found that asphalt concrete with the use of unmodified bitumen had a higher water saturation than asphalt concrete with the participation of a modified bitumen binder.

The introduction of additives in bitumen, which was used in the production of the asphalt concrete mixture, reduced water saturation and reached a minimum value at a dosage of 1 g/dm$^3$. At the same time, there was a close correlation in the patterns of change in the cosines of the contact angle of wetting in the binary system "bitumen-additive" and the amount of water absorbed by the samples in a given saturation mode. The largest decrease in *W*, as well as a decrease in cos θ, was recorded in the presence of a solution of polyisobutylene in mineral oil (AG-4I). With an increase in the concentration of AG-4I from 0 to 1 g/dm$^3$, the water saturation index decreased by 1.96 times and amounted to 4.02%, and for the other two additives, it was no more than 4.85–4.92% (Table 7).

### 4.2. Ternary Systems "Bitumen-AG-4I-AC-1"

It follows from the presented data that the introduction of AC-1 into bitumen compositions with a solution of polyisobutylene led to a deeper decrease in the specific surface energy $\sigma_{l\text{-}g}$ and, as a result, a denser packing of surfactant molecules on the surface. Compaction of the hydrophobic layer with the addition of AC-1 shifted the contact angle of wetting to the region of large values and reached a maximum at $C_{AG\text{-}4I}$ = 1 g/dm$^3$ and $C_{AC\text{-}1}$ = 1 g/dm$^3$. At these concentrations, the composition "bitumen—AG-4I—AC-1" provided an increase in the contact angle of wetting with water by 20.85° relative to unmod-

ified bitumen ($\theta = 95.05°$). For a comparative assessment of the water-repellent properties of additives in binary and ternary systems, we used a comparison of the indicators $\theta_{ex}$ and $\theta_c$.

The data obtained (Table 9) indicate that in the entire studied range of concentrations ($C$; $g/dm^3$: $0.5 \div 2.0$) there were deviations from the calculated data, except for the only ratio between the additives ($C_{AG-4I} = 1\ g/dm^3$ and $C_{AC-1} = 1\ g/dm^3$), for which the contact angles of wetting with water, $\theta_c = 115.46°$ and $\theta_{ex} = 115.90°$, had almost equal values ($\Delta\theta = -0.44$).

Beyond this ratio, the difference between the calculated and experimental wetting angles tended to increase ($\theta_c > \theta_{ex}$); however, in the region of increased concentrations of the sealing liquid ($C_{AG-4I} \geq 1.5\ g/dm^3$), on the contrary, the experimental values of the contact angle of wetting exceeded the calculated ones.

Thus, the maximum hydrophobic effect ($\theta = 105.90°$) was provided with the additive contribution of modifiers, i.e., in the absence of intermolecular interactions between surfactant molecules in the surface layer "bitumen-air".

The results of subsequent tests of asphalt concrete samples using bitumen containing AG-4I and AC-1 in their composition confirmed their decisive role in the development of water saturation processes (Figure 3).

With the simultaneous addition of two modifiers, the formation of a hydrophobic surfactant film on the bitumen surface reduced the water saturation of the modified samples with almost all variations in the quantitative contents of AG-4I and AC-1 (Figure 3). At the same time, the best indicator of water resistance (min W) was achieved with the same ratio of additives ($C_{AC-1} = 1\ g/dm^3$ and $C_{AG-4I} = 1\ g/dm^3$) (Table 12) as the best hydrophobicity index (min cos $\theta$, Figure 4). With this optimal content of two additives, the compacted hydrophobic film reduced the water saturation of the modified asphalt concrete sample by 2.6 times in comparison with the original bitumen and was 3.05%. This effect of water resistance was 1.3–1.6 times higher than with the individual use of additives.

**Table 12.** Water saturation index in the ternary systems "bitumen–AG-4I–AC-1".

| AG-4I, $g/dm^3$ | AC-1, $g/dm^3$ | W, % | AC-1, $g/dm^3$ | W, % | AC-1, $g/dm^3$ | W, % | AC-1, $g/dm^3$ | W, % | AC-1, $g/dm^3$ | W, % |
|---|---|---|---|---|---|---|---|---|---|---|
| 0 | 0 | 7.87 | 0.5 | 6.05 | 1.0 | 4.85 | 1.5 | 4.90 | 2.0 | 5.05 |
| 0.5 | 0 | 5.65 | 0.5 | 5.20 | 1.0 | 4.55 | 1.5 | 4.20 | 2.0 | 3.90 |
| 1.0 | 0 | 4.02 | 0.5 | 3.50 | 1.0 | 3.05 | 1.5 | 3.20 | 2.0 | 3.82 |
| 1.5 | 0 | 4.80 | 0.5 | 4.65 | 1.0 | 4.00 | 1.5 | 3.75 | 2.0 | 3.80 |
| 2.0 | 0 | 5.20 | 0.5 | 4.70 | 1.0 | 4.25 | 1.5 | 4.20 | 2.0 | 4.50 |

*4.3. Correlation between the Marginal Angle of Wetting Bitumen Films with Water and the Indicator of Water Saturation of Asphalt Concrete*

Partial dependencies (Figure 6) were approximated by one-parameter equations that were combined into multi-factor mathematical models (Equations (7) and (8)).

$$\theta = \frac{\left(-2.6166 C_{AC-1}^2 + 7.9655 C_{AC-1} + 103.57\right)\left(-5.2137 C_{AG-4I}^2 + 14.848 C_{AG-4I} + 100.58\right)}{107.612} \tag{7}$$

$$W = \frac{\left(0.6554 C_{AC-1}^2 - 1.9825 C_{AC-1} + 5.5457\right)\left(1.3406 C_{AG-4I}^2 - 3.2507 C_{AG-4I} + 5.7863\right)}{4.5464} \tag{8}$$

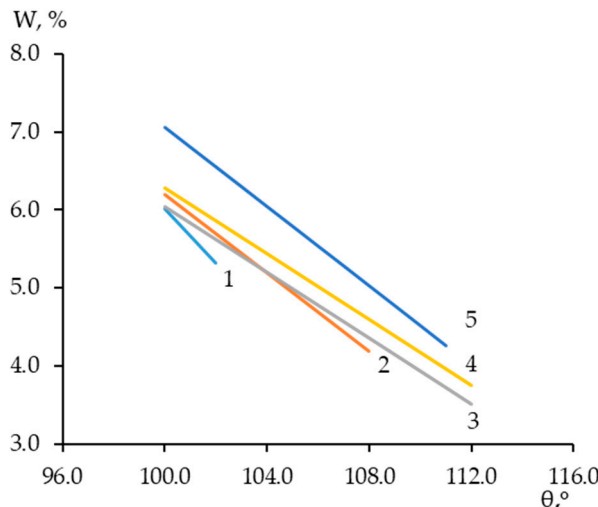

**Figure 6.** Nomogram of the dependence of the water saturation index of asphalt concrete on the contact angle of wetting with water of modified bitumen films. Concentration of AG-4I ($C_{AG-4I}$): 1, 0 g/dm³; 2, 0.5 g/dm³; 3, 1.0 g/dm³; 4, 1.5 g/dm³; 5, 2.0 g/dm³.

The calculated values, according to Equations (3) and (4), of coefficients of nonlinear multiple correlation (R > 0.89) and significance ($t_R$ > 2) indicate a satisfactory convergence of the experimental and calculated function values.

To find the correlation between the contact angle of wetting of bitumen films and the water saturation index, generalized Equations (7) and (8) were used. The contents of AC-1 additives were calculated according to Equation (7) at given values of the contact angle of wetting and AG-4I contents. By substituting the calculated values of the AC-1 content into Equation (8), we found the water saturation indicators, thereby forming the dependence W = f(θ) shown in Figure 6, which allowed for predicting and determining one of the strength characteristics of asphalt concrete: the water saturation parameter at a known value of the contact angle of wetting. Such an approach to determining the water saturation parameter makes it possible to significantly reduce the duration of studies of the structural characteristics of asphalt concrete by using a less labor-intensive and more rapid method, which is an automated method for determining the contact angle of wetting.

Generalized equations and a graphical representation of a function of several variables (Figure 6) allowed for optimizing compositions by the content of AG-4I and AC-1 modifiers to achieve the required performance properties of asphalt concrete coatings.

## 5. Conclusions

Based on the experimental research conducted, the following conclusions were reached:

1. Modifiers for the hydrophobic effect of the bituminous coating formed the series (in decreasing order): AG-4I > AC-1 > AMDOR-10. The maximum increase in the contact angle of wetting with water falls on their concentration of 1 g/dm³. In comparison with unmodified bitumen (θ = 95.05°), the wetting angle increased by 12.25° (AG-4I), by 8.16° (AC-1), and by 6.76° (AMDOR-10).

2. The hydrophobic effect of additives was determined by the degree of shielding of the bitumen surface by hydrophobic regions of surfactant molecules. This process was determined by the length, the branching of the hydrocarbon radical, and the number of hydrocarbon chains of additives. The concentration of surfactants at the "bitumen-air" interface showed a decrease in surface tension relative to unmodified bitumen (Δσ): 4.59 mN/m (AC-1), 4.08 mN/m (AG-4I), and 0.68 mN/m (AMDOR-10).

3. A close correlation was found between the hydrophobizing effect of modifiers in bitumen and the water resistance of modified asphalt concrete. The minimum water saturation of asphalt concrete samples was recorded at the same concentration of all

studied additives (C = 1 g/dm$^3$), at which the maximum contact angle of wetting with water θ (min cos θ) was observed.

4. With the content of modifiers 1 g/dm$^3$, the best results were achieved in the presence of AG-4I: water saturation decreased by 1.96 times (relative to the base variant without AG-4I) and amounted to 4.02%. The water saturation of asphalt concrete with the introduction of nitrogen-containing modifiers was at the levels of 4.85% (AC-1) and 4.92% (AMDOR-10).

5. The addition of AC-1 to the "bitumen-AG-4I" system shifted the contact angle of wetting to higher values, with a maximum at $C_{AG-4I}$ = 1 g/dm$^3$ and $C_{AS-1}$ = 1 g/dm$^3$. At these concentrations, the change in the contact angle of wetting by 20.85° in comparison to unmodified bitumen was an additive value ($θ_c$ = 115.46° and $θ_{ex}$ = 115.90°), which ensured maximum shielding of the solid surface from water.

6. The simultaneous presence of AG-4I and AC-1 in the binder at their optimal ratio ($C_{AG-4I}$ = 1 g/dm$^3$ and $C_{AC-1}$ = 1 g/dm$^3$) led to a deeper decrease in the water saturation of asphalt concrete samples W = 3.05% than that of their individual use.

7. A close correlation was revealed between the hydrophobicity of modified bitumen and the water saturation of asphalt concrete. Generalized equations and a graphical representation of a function of several variables allowed for optimizing compositions by the content of AG-4I and AC-1 modifiers to achieve the required performance properties of asphalt concrete coatings.

8. The additives we offer can be used as effective modifiers to increase the hydrophobicity of asphalt concrete pavements. The water repellency of the two modifiers AG-4I and AC-1 manifested itself at the maximum level when they were introduced simultaneously into bitumen-mineral compositions in the optimal ratio. Additive AC-1 is obtained from petrochemical waste, which is undoubtedly a cost-effective factor, and the use of waste sealing liquid AG-4I is expedient from the point of view of environmental protection.

**Author Contributions:** Conceptualization, A.D. and Y.B.; methodology, A.D. and Y.B.; software, K.O.; validation, A.D. and K.O.; formal analysis, A.L.; investigation, A.D. and Y.B.; resources, Y.B.; data curation, A.D.; writing—original draft preparation, A.D. and Y.B.; writing—review and editing, A.D. and Y.B.; visualization Y.B. and K.O.; supervision, A.D.; project administration, Y.B., A.D., and A.L.; funding acquisition, A.L. All authors have read and agreed to the published version of the manuscript.

**Funding:** This research was funded by the Science Committee of the Ministry of Science and Higher Education of the Republic of Kazakhstan (Grant No. AP19677707).

**Institutional Review Board Statement:** Not applicable.

**Informed Consent Statement:** Not applicable.

**Data Availability Statement:** The datasets generated and/or analyzed during the current study are available from the corresponding author upon reasonable request.

**Conflicts of Interest:** The authors declare no conflict of interest.

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
