# Peer review of "Correlation Dependence between Hydrophobicity of Modified Bitumen and Water Saturation of Asphalt Concrete"

_applsci, doi:10.3390/app131910946_

Round 1

Reviewer 1 Report

The efforts made by the authors to establish the effect of modifiers on the hydrophobicity of bituminous films in order to achieve minimum water saturation and build a mathematical model of the wetting process with water. Although the idea is good, the process is too long and is not clear to the reader. The English writing of this paper is poor. Authors must check the sentences in paper. A lot of sentences are confusing and not understandable. Numerous obvious mistakes can be found in paper. The analysis results and discussions are not sufficient. The methodology introduced in section 2.7 (Method of probabilistic-deterministic planning) is lengthy and confusing and it was not explained how to determine some of parameters. When dealing with different materials in pavement industry, we need to simplify the process so it can be easily adopted. I advise the authors to rewrite it and the main problems of the paper are listed as follows:

 1-     Line 13-16 in abstract in confusing according to the grammar and writing mistakes. The specification name and its details should be removed.

 2-     Line 15, what are AG-4I and AC-1?? Be more careful of using acronym throughout the manuscript.

 3-     Line 75-81, The sentence meaning is not clear and there are multiple grammar and writing mistakes which mislead the concept.

 4-     Line 83-84, what do the authors mean by saying …"secondary raw materials in the production of building and road building materials"…??? The words used for this sentence is not correct.

 5-     Line 86, what does it mean …" as a means of "…?? The paper should be readable and understandable.

 6-     Line 89, Again writing mistake, …” the purpose of these studies is"…?? Which modifiers?? What does it mean… "modifiers of various nature"…??? Why the modifiers’ names were not mentioned in objective?? The key objectives must be clear and concise and illustrate the innovation points.

 7-     The collected data was not well-organized due to lack of in-depth analysis for each part of data sets.

 8-     Line 413, what does the authors mean, by saying …"the joint introduction of additives"…???

 9-     The figures must be mentioned exactly after mentioning in the text. It is very difficult to follow the discussions about the results of Figures 4, 5, and 6.

 10-  what is the practical advantage of this model??? How could author consider findings of this study in pavement design and maintenance??

 11-  Section 4.3 is very complex and it is not clear how did equations (6) and (7) obtained from??? Where is the explanation about Figure 6?

 12-  In Figure 7, what are these numbers illustrate??  What does CАG-4 mean??? There are many parameters in the manuscript which need to be clearly explained.

 13-  The conclusion must be rewrite in an appropriate way. For example, in line 450, it is not clear what are the authors pointing out??? In conclusions, the key findings of study must be mentioned precisely and concisely, instead of repeating the analysis results.

 14-  There are many references related to polymers, especially SBS. What is the importance of citing these papers in literature, while the modifiers used in this study were not similar. 

The English writing of this paper is poor. A lot of sentences are confusing and not understandable. Numerous obvious mistakes can be found in paper.

Author Response

Response to Reviewer

The efforts made by the authors to establish the effect of modifiers on the hydrophobicity of bituminous films in order to achieve minimum water saturation and build a mathematical model of the wetting process with water. Although the idea is good, the process is too long and is not clear to the reader. The English writing of this paper. Authors must check the sentences in paper. A lot of sentences are confusing and not understandable. Numerous obvious mistakes can be found in paper. The analysis results and discussions are not sufficient. The methodology introduced in section 2.7 (Method of probabilistic-deterministic planning) is lengthy and confusing and it was not explained how to determine some of parameters. When dealing with different materials in pavement industry, we need to simplify the process so it can be easily adopted. I advise the authors to rewrite it and the main problems of the paper are listed as follows:

  • Line 13-16 in abstract in confusing according to the grammar and writing mistakes. The specification name and its details should be removed.

Response: corrected

  • Line 15, what are AG-4I and AC-1?? Be more careful of using acronym throughout the manuscript.

Response: detailed description added

  • Line 75-81, The sentence meaning is not clear and there are multiple grammar and writing mistakes which mislead the concept.

Response: text was edited

  • Line 83-84, what do the authors mean by saying …"secondary raw materials in the production of building and road building materials"…??? The words used for this sentence is not correct.

Response: text was edited

  • Line 86, what does it mean …" as a means of "…?? The paper should be readable and understandable.

Response: text was edited

  • Line 89, Again writing mistake, …” the purpose of these studies is"…?? Which modifiers?? What does it mean… "modifiers of various nature"…??? Why the modifiers’ names were not mentioned in objective?? The key objectives must be clear and concise and illustrate the innovation points.

Response: description of studied additives was added in abstract

  • The collected data was not well-organized due to lack of in-depth analysis for each part of data sets.

Response: we have revised the text of paper (corrections lighted by yellow color)

  • Line 413, what does the authors mean, by saying …"the joint introduction of additives"…???

Response: we meant ‘Simultaneous addition of two modifiers’

  • The figures must be mentioned exactly after mentioning in the text. It is very difficult to follow the discussions about the results of Figures 4, 5, and 6.

Response: corrected

  • what is the practical advantage of this model??? How could author consider findings of this study in pavement design and maintenance??

Response: Generalized equations and a graphical representation of a function of several variables (Figure 6) allow optimizing compositions by the content of AG-4I and AC-1 modifiers to achieve the required performance properties of asphalt concrete coatings. Such an approach to determining the water saturation parameter makes it possible to significantly reduce the duration of studies of the structural characteristics of asphalt concrete by using a less labor-intensive and more rapid method, which is an automated method for determining the contact angle of wetting.

  • Section 4.3 is very complex and it is not clear how did equations (6) and (7) obtained from??? Where is the explanation about Figure 6?

Response: corrected

  • In Figure 7, what are these numbers illustrate?? What does CАG-4 mean??? There are many parameters in the manuscript which need to be clearly explained.

Response: we replaced Figure 7 (now it is the Figure 6) to the ‘Discussion’ and added explanation for CАG-4I

  • The conclusion must be rewrite in an appropriate way. For example, in line 450, it is not clear what are the authors pointing out??? In conclusions, the key findings of study must be mentioned precisely and concisely, instead of repeating the analysis results.

Response: Conclusions were rewritten.

  • There are many references related to polymers, especially SBS. What is the importance of citing these papers in literature, while the modifiers used in this study were not similar.

Response: Due to the fact that SBS is the most commonly used polymer modifier for asphalt concrete, it is most often mentioned in the literature review. The additives used by us can also be attributed to polymers of different molecular weights. So mentioning of SBS we considered to be relevant.

Reviewer 2 Report

1. Figures 6 and 7 show two and five curves, respectively. The authors could have created a nice legend and made it easier to understand which Multilevel plan-mixture they belong to, but the numbers on the diagram cannot be clearly distinguished like this.

2. A comparison was made with only one paper in the Discussion chapter. Has anyone else done some approximate tests where it would be possible to look back at other authors, compare the results with your results, and briefly describe the deviations of your results compared to other authors?

3. Conclusions are all nicely written, but it would be nice to put one sentence like "Based on the experimental research done, the following conclusions are reached:". This way, when the enumeration starts immediately, it needs to be clarified what it is about, so you should make a small introduction with a sentence.

Author Response

Response to Reviewer

  1. Figures 6 and 7 show two and five curves, respectively. The authors could have created a nice legend and made it easier to understand which Multilevel plan-mixture they belong to, but the numbers on the diagram cannot be clearly distinguished like this.

Response: corrected

  1. A comparison was made with only one paper in the Discussion chapter. Has anyone else done some approximate tests where it would be possible to look back at other authors, compare the results with your results, and briefly describe the deviations of your results compared to other authors?

In contrast to the mentioned studies, we use oil refining waste as modifiers for the synthesis of additive, and the second one is a spent polymer sealant. Based on the results of the literature review, we can conclude that no similar studies have been found related specifically to the establishment of a correlation between the indicators of hydrophobicity of modified bitumen and water saturation of asphalt concrete.

  1. Conclusions are all nicely written, but it would be nice to put one sentence like "Based on the experimental research done, the following conclusions are reached:". This way, when the enumeration starts immediately, it needs to be clarified what it is about, so you should make a small introduction with a sentence.

Response: corrected

Reviewer 3 Report

Dear Authors,

 The aim of this paper is to improve the durability of asphalt concrete road surfaces by increasing their moisture resistance. The idea is interesting and could be exploited in the future.  I welcome the idea of these studies which are trying to establish the effect of modifiers of various nature on the hydrophobicity of bituminous films in order to achieve minimum water saturation and build a mathematical model of the wetting process with water.

 As week elements

I understood how to obtain the values of the answers on a line as the arithmetic mean of the 5 answers on the same line, what y1-y25 represent and how they were obtained (table 4). And what is the combination of the two parameters that gives us these results?! The request is to write this matrix explicitly!

In Figures 4 - 7, the sizes on the two reference axes are not clear - they should be written explicitly, not just with initials, and of course I suggest writing them in the middle, not at the end of the axis. How the two equations 5 and 6 were obtained based on the curves in figure 6?! It is not very clear.

None of the described modifiers are presented further! I would have liked their physico-chemical properties to be presented, for some chemical formulas to appear, perhaps most importantly, if there are any chemical reactions between them and certain compounds present in the composition of the bitumen!

Maybe it would have been interesting to do some SEMs or TEMs or other types of chemical analysis to demonstrate the degree of homogeneity of the mixtures and especially their stability (polymers tend to separate from bitumen, with the passage of time)! Perhaps in a future work the authors could present such information!

 As notable elements

With over 50 bibliographic references, with experimental tests appropriate to the research carried out but also with o lot of mechanical tests, the authors managed to demonstrate that the proposed solution is a relevant one.

It can be noted that the mathematical apparatus has been well used to perform the necessary demonstrations!

Author Response

Response to Reviewer

 As weak elements

I understood how to obtain the values of the answers on a line as the arithmetic mean of the 5 answers on the same line, what y1-y25 represent and how they were obtained (table 4). And what is the combination of the two parameters that gives us these results?! The request is to write this matrix explicitly!

Response: y1-y25 are response functions of 25 experiments. We rewrite paper to make it more clear.

In Figures 4 - 7, the sizes on the two reference axes are not clear - they should be written explicitly, not just with initials, and of course I suggest writing them in the middle, not at the end of the axis. How the two equations 5 and 6 were obtained based on the curves in figure 6?! It is not very clear.

Response: We believe that the way we named axis is more representative, so we would like to leave it as it is. We rewrite paper to make it more clear how equations 5 and 6 were obtained.

None of the described modifiers are presented further! I would have liked their physico-chemical properties to be presented, for some chemical formulas to appear, perhaps most importantly, if there are any chemical reactions between them and certain compounds present in the composition of the bitumen!

Response: AC-1 was synthesized at our university, so a more complete description is given for it. AG-4I and AMDOR-10 are industrial products, so their composition and properties were described limitedly. According to results of our experiments no chemical reactions between components of all studied systems were identified.

Maybe it would have been interesting to do some SEMs or TEMs or other types of chemical analysis to demonstrate the degree of homogeneity of the mixtures and especially their stability (polymers tend to separate from bitumen, with the passage of time)! Perhaps in a future work the authors could present such information!

Response: Thank you for your comments! In our further researches we will study this effect.

 As notable elements

With over 50 bibliographic references, with experimental tests appropriate to the research carried out but also with o lot of mechanical tests, the authors managed to demonstrate that the proposed solution is a relevant one.

It can be noted that the mathematical apparatus has been well used to perform the necessary demonstrations!

Round 2

Reviewer 1 Report

The authors replied to the comments accurately and the paper has been revised accordingly. Therefore, it can be accepted for publication in the current form.